# Deep Learning-Based Depression Detection from Social Media: Comparative Evaluation of ML and Transformer Techniques

Biodoumoye George Bokolo * and Qingzhong Liu

Department of Computer Science, Sam Houston State University, Huntsville, TX 77341, USA; qxl005@shsu.edu
* Correspondence: bgb023@shsu.edu

**Abstract:** Detecting depression from user-generated content on social media platforms has garnered significant attention due to its potential for the early identification and monitoring of mental health issues. This paper presents a comprehensive approach for depression detection from user tweets using machine learning techniques. The study utilizes a dataset of 632,000 tweets and employs data preprocessing, feature selection, and model training with logistic regression, Bernoulli Naive Bayes, random forests, DistilBERT, SqueezeBERT, DeBERTA, and RoBERTa models. Evaluation metrics such as accuracy, precision, recall, and F1 score are employed to assess the models' performance. The results indicate that the RoBERTa model achieves the highest accuracy ratio of 0.981 and the highest mean accuracy of 0.97 (across 10 cross-validation folds) in detecting depression from tweets. This research demonstrates the effectiveness of machine learning and advanced transformer-based models in leveraging social media data for mental health analysis. The findings offer valuable insights into the potential for early detection and monitoring of depression using online platforms, contributing to the growing field of mental health analysis based on user-generated content.

**Keywords:** depression detection; social media analysis; deep learning models; NLP techniques; user tweets; mental health identification; sentiment analysis; large language models





## 1. Introduction

### 1.1. Background

Depression is a prevalent mental health condition that affects a substantial number of individuals worldwide [1]. It is characterized by persistent feelings of sadness, loss of interest, and impaired functioning, leading to a significant decline in overall well-being and quality of life [2]. Timely detection and intervention are crucial for the effective management and treatment of depression. Left untreated, depression can lead to severe impairments in personal, social, and occupational functioning [3].

The advent of social media platforms has provided an unprecedented opportunity to study mental health conditions, including depression, on a large scale [4]. Twitter, in particular, has emerged as a valuable source of data for understanding individuals' thoughts and emotions. Twitter users often openly share their personal experiences, feelings, and emotions, making it possible to explore and identify signs of depression through their public messages [5].

Machine learning techniques have demonstrated immense potential in automatically analyzing vast volumes of textual data and extracting meaningful insights [6,7]. Natural language processing (NLP) algorithms, in particular, have been leveraged to develop computational models capable of detecting depression symptoms in user-generated content, such as tweets [8]. These models offer a promising avenue to complement traditional diagnostic approaches and provide an efficient, scalable, and cost-effective means of screening for depression at a large scale [9,10].

*1.2. Objective of the Study*

The objective of this study is to investigate the feasibility of detecting depression from user tweets using machine learning techniques. By analyzing a dataset comprising Twitter posts, we aim to develop a robust and accurate predictive model capable of identifying users who may be at risk of depression. This research can contribute to the development of innovative digital health tools that facilitate early intervention and provide support to individuals in need of mental health assistance.

*1.3. Research Gap*

Financial and time constraints often limit the creation of dedicated datasets for depression detection. The study proposes to bridge the gap by repurposing an existing dataset originally designed for sentiment analysis and adapting it for depression detection using custom algorithms that are cost-effective and align with real-world resource constraints. In addition, the existing literature often lacks comprehensive comparative analyses of a combination of traditional machine learning and transformer models for depression detection. This study addresses this gap by evaluating the performance of multiple deep learning models, including RoBERTa, SqueezeBERT, DeBERTA, DistilBERT, logistic regression, Naive Bayes, and random forests, and comparing their effectiveness in identifying depressive content.

In this paper, we present an overview of related work in the field of depression detection from social media data, emphasizing the application of machine learning algorithms. We describe the methodology employed in our study, including data collection, preprocessing, and feature engineering techniques. Furthermore, we discuss the features used to represent the textual content of tweets and the selection of appropriate machine learning algorithms for classification. Finally, we present the experimental results, evaluate the performance of all models used, and discuss potential applications and future directions for this research.

## 2. Literature Review

In recent years, there has been a growing interest in the use of artificial intelligence (AI) and machine learning (ML) to improve mental health care. As Shikha et al. (2023) [11] discussed, AI and ML can be used to detect and diagnose mental health conditions, develop AI-powered interventions, and improve access to mental health care services.

There has been a growing body of research exploring the detection of depression from social media data [12], particularly utilizing machine learning techniques. This section provides an overview of key studies and methodologies in the field, highlighting the advancements made in detecting depression through user tweets.

Negative comments or expressions of pessimism are often associated with depressive tendencies [13]. Research studies have explored the link between negative language use and depression, providing evidence to support the statement [9].

In a study conducted by [5], the researchers analyzed social media data and found a significant correlation between the language used in tweets and the prevalence of depression symptoms. They identified that individuals with higher levels of depression were more likely to express negative sentiments in their tweets.

In another study, [9] investigated the association between language markers and depression on social media platforms. They found that individuals with depressive symptoms tended to use more negative language, indicating a correlation between negative expression and depression.

Gkotsis et al. [14] employed informed deep learning techniques to characterize mental health conditions in social media. They utilized a large-scale dataset of Twitter posts and applied deep learning algorithms to detect mental health conditions, including depression. Their approach showcased the potential of leveraging deep learning models to gain insights from user-generated content and improve mental health monitoring.

Moreover, the study by Resnik et al. [15] explored the role of sentiment analysis and linguistic markers in detecting depression from Twitter data. They developed a machine learning framework that incorporated sentiment analysis features to predict depression levels in individuals. Their findings highlighted the importance of sentiment analysis in capturing emotional states and identifying signs of depression.

The study [16] focused on detecting depression using social media data and machine learning employed various text classification algorithms, including Support Vector Machines (SVMs) and random forests, to classify tweets as depressive or non-depressive. As explained by Kim (2017) [17], SVMs work by finding a hyperplane in the data that separates the two classes (depressed vs. not depressed) with the maximum margin. The study achieved promising results in terms of classification accuracy, demonstrating the potential of machine learning approaches for depression detection. While the study demonstrated effective depression detection from social media data, it primarily focused on traditional machine learning algorithms. Incorporating more advanced deep learning models such as recurrent neural networks or transformers could potentially improve the performance and capture complex patterns within the tweet data.

In [18], the researchers examined the use of natural language processing techniques to analyze social media posts for detecting depression. They applied sentiment analysis and topic modeling to identify linguistic markers associated with depression. The study highlighted the importance of linguistic cues in identifying mental health conditions from social media data. Although the study provided valuable insights into the linguistic markers of depression, the research focused solely on Twitter activity and did not explore the potential of utilizing additional contextual information from user profiles or network interactions. Integrating these additional features could enhance the accuracy and robustness of depression detection models.

Guntuku et al. (2017) [19] investigated the relationship between language patterns on Twitter and depression symptoms. Their research explored the differences in linguistic style, linguistic content, and social engagement between depressed and non-depressed individuals. The study highlighted the importance of considering social context and interaction patterns in depression detection. The study provided a comprehensive review of the existing literature on detecting depression from social media. However, further research is needed to explore the generalizability of the findings to diverse populations and different social media platforms, as user behavior and language use may vary across platforms and cultural contexts.

Other studies have also emphasized the significance of incorporating contextual information from social media platforms [20]. For example, Nguyen et al. [21] investigated the relationship between social context and depression detection, analyzing not only the content of tweets but also the social network connections between users. Their research highlighted the potential benefits of considering the social context in understanding mental health indicators on social media.

Overall, these studies collectively demonstrate the potential of using machine learning techniques for detecting depression from user tweets. By analyzing linguistic patterns, social interactions, and contextual information, researchers have made strides in developing computational models capable of identifying individuals at risk of depression.

## 3. Materials and Methods

This section outlines the methodology employed in this study to detect depression from user tweets using machine learning techniques. The methodology encompasses data collection, preprocessing, feature extraction, and the selection and training of machine learning algorithms.

### 3.1. About the Dataset

Tweets could be obtained from the Twitter website using the Twitter developer API with a monthly or yearly subscription fee. However, due to financial and time constraints,

we utilized an already existing dataset, the Sentiment140 dataset, which was first used in the journal article by Go, Richa, and Lei (2009) [22] and originally intended for sentiment analysis. For the purpose of this study on depression detection, we applied a custom algorithm to repurpose and re-label the dataset, as described in Section 3.2. The Sentiment140 dataset consists of approximately 1.6 million English-language tweets collected from Twitter. These tweets were gathered from April to June 2009 and covered a wide range of topics and sentiments expressed by users. Each tweet in the dataset is labeled with sentiment polarity, specifically as either positive or negative, as can be seen in Table 1.

**Table 1.** Description of the Sentiment140 dataset.

| Feature | Details |
| --- | --- |
| Target | The sentiment of the tweet (where 0 = negative, 2 = neutral, 4 = positive) |
| ID | The ID of the tweet (e.g., "2087") |
| Date | The date of the tweet (e.g., "Sat May 16 23:58:44 UTC 2009") |
| UserID | The user that tweeted (e.g., "coolboy21") |
| Text | The text of the tweet (e.g., "James is cool") |

*3.2. Dataset Transformation and Relabeling*

We acknowledge that the Sentiment140 dataset contains tweets labeled as either positive or negative sentiment. To adapt this dataset for depression detection, we inferred the presence of depression from tweets with a negative sentiment. Studies suggest that negative comments or language use can be indicative of a predisposition toward depression [5,9]. By monitoring language patterns in social media, such as those captured in the Sentiment140 dataset, the system can potentially identify early warning signs or indicators of depression [23]. Specifically, we employed a custom algorithm that analyzed the textual content of each tweet.

1. Average Polarity Calculation: The polarity score of a sentence or word is a measure of how positive or negative the sentiment of the word or sentence is (between $-1$ and $+1$). To distinguish between depressive and non-depressive tweets among those classified as having a negative sentiment, we computed the average polarity score for a set of words highly correlated with depression. These words were identified based on the premise noted by Yazdavar et al. (2017) that by monitoring language patterns in social media, such as those captured in the Sentiment140 dataset, the system can potentially identify early warning signs or indicators of depression. This established their relevance to depression detection. The code for finding this polarity is shown in Figure 1.

2. Tweet Categorization: With the average polarity score as a threshold, we disregarded the sentiment labels and categorized the tweets (code shown in Figure 2) as follows:

   - Non-Depressive Tweets: Tweets with a polarity score higher than the threshold were re-labeled as non-depressive tweets. Some of these tweets may have been considered indicative of a negative sentiment but not necessarily reflective of depression.
   - Depressive Tweets: Tweets with a polarity score lower than the threshold were re-labeled as depressive tweets. These tweets were inferred to have a higher likelihood of containing depressive content.
     By employing this custom algorithm, we repurposed the Sentiment140 dataset for depression detection; the system can extract valuable insights regarding the users' mental well-being [5,9]. Specifically, the system will focus on identifying negative expressions or comments, which have been shown to correlate with depressive symptoms [5,9].

```python
polarity = []
# Loop through each word in the 'words' list
for word in words:
    text = TextBlob(word)
    # Calculate the polarity of the word using TextBlob
    polarity.append(text.sentiment.polarity)
# get the mean polarity
sep_polarity = mean(polarity)
```

**Figure 1.** Code to find the average polarity of tweets.

```python
def get_label(tweet):
    """
    Function to determine the label of a tweet based on its sentiment polarity.

    Parameters:
    - tweet (str): The text of the tweet

    Returns:
    - label (str): The label of the tweet ('NonDepressed' or 'Depressed')
    """
    tweet = TextBlob(tweet)
    sentiment = tweet.sentiment.polarity
    if sentiment > sep_polarity:
        return 'NonDepressed'
    return 'Depressed'
```

**Figure 2.** Code to compute the depressive label of each tweet.

It is important to note that while this adaptation of the dataset with the description shown in Table 2 provides a starting point for depression detection, it may not capture all nuances of depression, and further refinement and validation may be necessary, some of which are not covered in this study.

**Table 2.** Description of the **transformed** dataset.

| Feature | Details |
| --- | --- |
| Label | Label showing whether the tweet is showing depressive symptoms or not. |
| Tweet | The text of the user's twitter post. |

After the transformation and re-labeling of the dataset, a word cloud diagram was used to analyze the new labels. The word cloud diagrams, shown in Figures 3 and 4, were used to visually compare the most frequently occurring words in depressive and non-depressive tweets, with the most frequently occurring words visually appearing larger than the less occurring words.

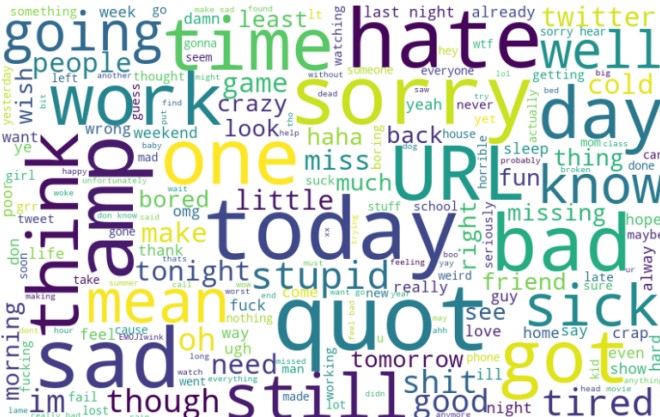

**Figure 3.** Word cloud showing frequency of depressive words.

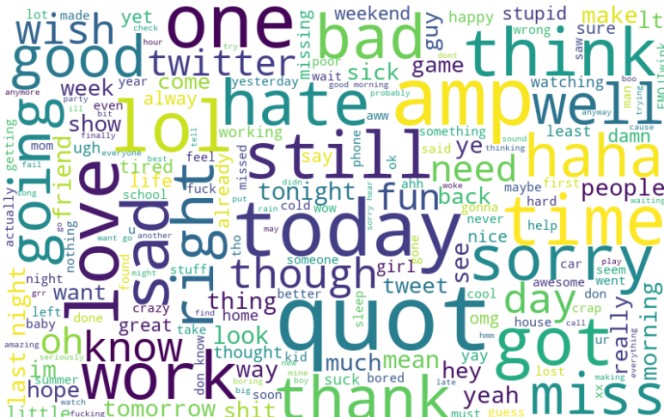

**Figure 4.** Word cloud showing frequency of non-depressive words.

*3.3. Data Preprocessing*

As part of the preprocessing step, we performed feature selection to identify and remove irrelevant or redundant features. In this study, the following features ('ids', 'date', 'flag', 'user') were deemed irrelevant to the task of depression detection from tweets and were dropped from the dataset.

1. Handling Missing Values: To ensure the quality and integrity of the dataset, we assessed the presence of missing values. Fortunately, no missing values were detected in the tweet data. This allowed us to proceed with the preprocessing steps without imputation or removal of incomplete instances.
2. Text Cleaning: To prepare the textual content of the tweets, we applied a series of cleaning operations:

   - Convert to Lowercase: We converted all text to lowercase to ensure consistency and avoid duplication caused by variations in capitalization.
   - Removing Links: We removed any URLs or hyperlinks present in the tweets, as they do not contribute to the semantic meaning and can introduce noise in the analysis.
   - Remove User Mentions: We eliminated user mentions (e.g., "@username") from the tweets, as they typically do not carry substantial sentiment information and can be treated as noise in the analysis.
   - Remove Non-Letters: We removed any non-alphabetic characters, such as numbers or special characters, as they generally do not contribute to the sentiment or meaningful analysis.

- Removing Stop Words: Stop words, such as "and", "the", or "is", do not typically carry significant sentiment information. Hence, we removed them from the tweet text to reduce noise and improve the accuracy of the analysis.
- Apply Stemming and Lemmatization: We performed stemming and lemmatization to reduce words to their root form. Stemming reduces words to their base or stem (e.g., "running" to "run"), while lemmatization aims to convert words to their base form based on their dictionary meaning (e.g., "better" to "good"). These techniques help standardize the text and reduce the dimensionality of the dataset. By applying these cleaning operations, we ensured that the tweet data were normalized, free of noise, and ready for subsequent feature extraction and machine learning tasks.

### 3.4. Feature Engineering

To prepare the target variable for machine learning algorithms, we performed label encoding [24]. The labels indicating depression and non-depression were encoded as numeric values. For instance, we assigned the value 1 to tweets indicating depression and 0 to tweets indicating non-depression. Label encoding allowed us to represent the target variable in a suitable format for training and evaluation.

Different machine learning models require input data in specific formats [25]. In this study, we employed the TF-IDF (Term Frequency-Inverse Document Frequency) vectorization technique to transform the textual tweet data into numerical representations suitable for specific models. TF-IDF represents each tweet as a numerical vector that considers the importance of words in the tweet and across the entire dataset [26]. It calculates the term frequency (TF) of each word in a tweet and scales it by the word's inverse document frequency (IDF) across the dataset. TF-IDF vectorization is widely used in models such as SVM and XGBoost.

### 3.5. Model Training

In this study, we conducted a comparative analysis of various natural language processing models using the framework shown in Figure 5 to evaluate their effectiveness in accurately classifying depression in Twitter posts. We compared the outcomes generated by these models and assessed their classification accuracy. The Sentiment140 dataset is used in conjunction with a corpus of words correlated to depression to generate the depression dataset, and then the dataset undergoes further preprocessing like lemmatization, stop word removal, and so on. After that, some feature engineering is completed to make the data available for use by the ML algorithms. For the transformer models (RoBERTa, DistilBERT, DeBERTA, and SqueezeBERT), label encoding is the only feature engineering performed on the data before training and evaluation. The traditional ML algorithms (logistic regression, Naive Bayes, and random forests) have to undergo both label encoding and vectorization before training and evaluation.

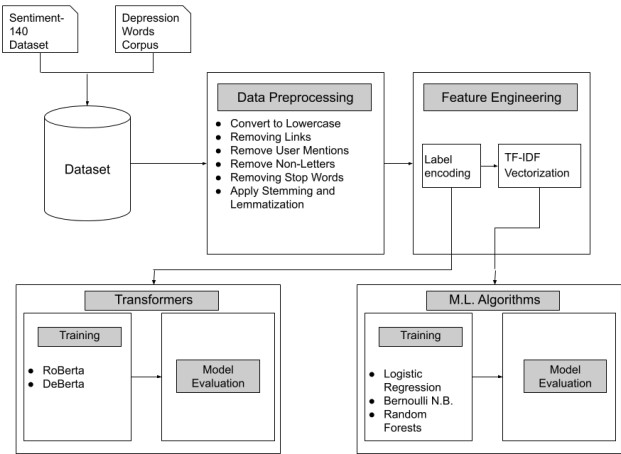

**Figure 5.** Proposed framework.

### 3.6. TextData Class for Data Loading and Preprocessing

In the context of deep learning-based text classification tasks, a well-structured data pipeline is crucial for efficiently loading and preprocessing the dataset. We have implemented a `TextData` class to facilitate this process, which extends the `Dataset` class from the PyTorch library. This class is designed to seamlessly prepare text data for use with transformer models.

#### 3.6.1. Key Features

- Initialization: The `TextData` class takes as input a dataframe containing text data, a tokenizer (typically from a transformer model), and the maximum sequence length (`max_len`) allowed for the input text.
- Data Loading: The `__init__` method in Figure 6 initializes the class by storing the tokenizer, dataframe, text data, labels (targets), and maximum sequence length.

```python
def __init__(self, dataframe, tokenizer, max_len):
    self.tokenizer = tokenizer
    self.data = dataframe
    self.text = dataframe.text
    self.targets = self.data.labels
    self.max_len = max_len
```

**Figure 6.** Data initialization code.

- Data Length: The `__len__` method shown in Figure 7 returns the total number of samples in the dataset, which is equivalent to the number of text entries.

```python
def __len__(self):
    return len(self.text)
```

**Figure 7.** Data length code.

- Data Retrieval: The `__getitem__` method shown in Figure 8 processes individual samples. It tokenizes the text using the tokenizer, ensuring that it adheres to the specified maximum length. The tokenized components, including input IDs, attention mask, and token type IDs, are extracted and returned along with the corresponding target labels.

```python
def __getitem__(self, index):
    text = str(self.text[index])
    text = " ".join(text.split())

    # Tokenize and preprocess the text using the provided tokenizer
    inputs = self.tokenizer.encode_plus(
        text,
        None,
        add_special_tokens=True,
        max_length=self.max_len,
        truncation=True,
        padding='max_length',
        return_token_type_ids=True
    )

    # Extract relevant input components
    ids = inputs['input_ids']
    mask = inputs['attention_mask']
    token_type_ids = inputs["token_type_ids"]
```

**Figure 8.** Code for data retrieval.

- __getitem__ return: The __getitem__ method shown in Figure 9 returns a dictionary containing the following elements for each data sample. This dictionary allows for easy access to the processed data when training deep learning models. For example, you can retrieve the input IDs using sample['ids'] or the target labels using sample['targets'], where sample is an instance of the dataset generated by the TextData class.

```
return {
    'ids': torch.tensor(ids, dtype=torch.long),
    'mask': torch.tensor(mask, dtype=torch.long),
    'token_type_ids': torch.tensor(token_type_ids, dtype=torch.long),
    'targets': torch.tensor(self.targets[index], dtype=torch.float)
}
```

**Figure 9.** What the getitem() method returns.

3.6.2. Traditional ML Algorithms

1. Logistic regression is a widely used linear classifier [27]. It models the relationship between the independent variables and the probability of a binary outcome, making it suitable for sentiment classification tasks. We trained a logistic regression model on the preprocessed tweet data to classify tweets as either indicative of depression or not. The hyperparameters used in training are C = 3, multi_class='ovr', and solver='liblinear'.
2. Bernoulli Naive Bayes [28] is a probabilistic classifier that assumes features are conditionally independent given the class variable. We utilized the Bernoulli Naive Bayes algorithm to capture the binary presence or absence of features in the tweet data for depression detection. Default hyperparameters were utilized for this algorithm.
3. Random forests [29] are ensemble learning algorithms that combine multiple decision trees to make predictions. They are capable of capturing complex interactions and non-linear relationships in the data. We employed random forests to leverage the ensemble of decision trees for depression classification based on the tweet features. The two hyperparameters used during training are n_estimators = 30, and n_jobs = −1.

*3.7. Transformer Model Fine Tuning*

1. RoBERTa [30] and DeBERTa [31] are transformer-based models that have achieved state-of-the-art performance in various natural language processing tasks. These models employ a transformer architecture that utilizes self-attention mechanisms to capture contextual relationships between words. We fine tuned pre-trained RoBERTa and DeBERTa models with the model architecture shown in Figure 10 on our dataset to leverage their advanced language modeling capabilities for depression detection from tweets.
2. DistilBERT is a transformer-based language model that has gained significant attention in natural language processing tasks [32]. In our study, we fine tuned the DistilBERT model with the model architecture shown in Figure 11 as part of our methodology for detecting depression through social media. DistilBERT, a distilled version of the BERT model, offers a more compact architecture while retaining its powerful language representation capabilities.
3. SqueezeBERT is another state-of-the-art pre-trained language model specifically designed for efficient computation on resource-constrained devices [33]. It employs a compact architecture that significantly reduces the model size while preserving the model's performance. It was fine tuned for this study using the model architecture shown in Figure 12.

```python
class DebertaClass(torch.nn.Module):
    def __init__(self):
        super(DebertaClass, self).__init__()
        self.l1 = DebertaModel.from_pretrained("microsoft/deberta-base")
        self.pre_classifier = torch.nn.Linear(768, 768)
        self.dropout = torch.nn.Dropout(0.3)
        self.classifier = torch.nn.Linear(768, 2)

    def forward(self, input_ids, attention_mask, token_type_ids):
        output_1 = self.l1(input_ids=input_ids, attention_mask=attention_mask,
         token_type_ids=token_type_ids)
        hidden_state = output_1[0]
        pooler = hidden_state[:, 0]
        pooler = self.pre_classifier(pooler)
        pooler = torch.nn.ReLU()(pooler)
        pooler = self.dropout(pooler)
        output = self.classifier(pooler)
        return output
```

**Figure 10.** Model architecture used for fine tuning of the DeBERTa model.

```python
class DistilBertClass(torch.nn.Module):
    def __init__(self):
        super(DistilBertClass, self).__init__()
        self.l1 = DistilBertModel.from_pretrained("distilbert-base-uncased")
        self.pre_classifier = torch.nn.Linear(768, 768)
        self.dropout = torch.nn.Dropout(0.3)
        self.classifier = torch.nn.Linear(768, 2)

    def forward(self, input_ids, attention_mask):
        output_1 = self.l1(input_ids=input_ids, attention_mask=attention_mask)
        hidden_state = output_1[0]
        pooler = hidden_state[:, 0]
        pooler = self.pre_classifier(pooler)
        pooler = torch.nn.ReLU()(pooler)
        pooler = self.dropout(pooler)
        output = self.classifier(pooler)
        return output
```

**Figure 11.** Model architechture used for fine tuning of DistilBERT model.

```python
class SqueezeBertClass(torch.nn.Module):
    def __init__(self):
        super(SqueezeBertClass, self).__init__()
        self.l1 = SqueezeBertModel.from_pretrained("squeezebert/squeezebert-uncased")
        self.pre_classifier = torch.nn.Linear(768, 768)
        self.dropout = torch.nn.Dropout(0.3)
        self.classifier = torch.nn.Linear(768, 2)

    def forward(self, input_ids, attention_mask, token_type_ids):
        output_1 = self.l1(input_ids=input_ids, attention_mask=attention_mask, token_type_ids=token_type_ids)
        hidden_state = output_1[0]
        pooler = hidden_state[:, 0]
        pooler = self.pre_classifier(pooler)
        pooler = torch.nn.ReLU()(pooler)
        pooler = self.dropout(pooler)
        output = self.classifier(pooler)
        return output
```

**Figure 12.** Model architechture used for fine tuning of SqueezeBERT model.

Loss Function and Optimizer

- Loss Function used: *Cross-Entropy Loss*
  The cross-entropy loss function is a commonly used objective function in classification tasks, including depression detection. It measures the dissimilarity between the

predicted probability distribution and the true labels [34]. In this study, the cross-entropy loss function was utilized to compute the loss during the training process [35]. The cross-entropy loss encourages the model to assign higher probabilities to the correct class and lower probabilities to the incorrect classes. By minimizing this loss, the model learns to make accurate predictions and capture the underlying patterns in the data.

- Optimizer used: *Adam Optimizer*
  Adam combined the benefits of two further stochastic gradient descent modifications. In particular:

  – The adaptive gradient algorithm, or AdaGrad, enhances performance on problems with sparse gradients (such as computer vision and natural language issues) by maintaining a per-parameter learning rate.

  – Root Mean Square Propagation (RMSProp), which also sustains per-parameter learning rates adjusted according to the mean of the weight's gradients' recent magnitudes (i.e., the rate of change). This indicates that the algorithm performs well in non-stationary, online scenarios. [36].

In this study, we employed an Adam optimizer, allowing for faster convergence and better performance compared to traditional optimization methods. It efficiently updated the model parameters during training, adjusting the learning rate based on the gradient magnitudes and past gradients.

By utilizing these models, we aimed to explore different approaches to detecting depression from user tweets. Each model has its strengths and characteristics, allowing us to compare their performance and determine the most effective approach for our specific task.

## 4. Results

The methodology used to evaluate the performance of the models for depression detection from user tweets is outlined here. The evaluation employed common classification metrics, including accuracy, precision, recall, F1 score, and confusion matrix diagram.

1. **Accuracy** measures the proportion of correctly classified instances out of the total instances. It is a fundamental metric that provides an overall assessment of the model's performance. In depression detection, it indicates the model's ability to correctly identify both depressive and non-depressive tweets, offering a clear picture of its general effectiveness.

2. **The F1 score** is the harmonic mean of precision and recall, both of which measure the proportion of prediction against actual instances. It provides a balanced measure that considers both false positives and false negatives. In depression detection, the F1 score helps strike a balance between minimizing both types of errors. A high F1 score indicates a model that effectively identifies depressive tweets while maintaining a low rate of misclassification.

3. **Precision** measures the proportion of true positive predictions out of all positive predictions made by the model. In the context of depression detection, precision signifies the model's ability to correctly identify tweets as depressive without making many false positive predictions. This is crucial, as misclassifying non-depressive tweets as depressive could have negative consequences.

4. **Recall**, also known as sensitivity or true positive rate, measures the proportion of true positive predictions out of all actual positive instances. In depression detection, recall indicates how well the model captures all the depressive tweets present in the dataset. It is particularly important because missing depressive tweets (false negatives) can be as problematic as false positives.

5. **The confusion matrix diagram** provides a detailed breakdown of the model's predictions, including true positives, true negatives, false positives, and false negatives. It offers insights into the distribution of correct and incorrect predictions, helping researchers understand where the model excels and where it may need improvement.

Visualizing the confusion matrix can also aid in identifying specific areas of concern and fine tuning the model accordingly.

### 4.1. Test Split and Cross-Validation

To ensure an unbiased evaluation of the models, the dataset of 632,000 rows was split into two subsets: a training set and an evaluation set. The splitting ratio chosen was 80% for training and 20% for evaluation. This means that approximately 80% of the dataset, which corresponds to 505,600 rows, was allocated for training the models. The remaining 20%, amounting to 126,400 rows, was reserved for evaluating the models' performance.

To ensure robustness and reliable performance assessment, all machine learning models employed in this study underwent a rigorous cross-validation process with 10 splits using 100,000 entries for traditional ML models and 50,000 entries for the transformer models. Each split involved dividing the dataset into 80% for training and 20% for validation. This approach allowed us to mitigate the potential impact of data partitioning on model performance and effectively evaluate the models' generalizability.

Furthermore, to provide a comprehensive summary of the model performance, we calculated the mean and standard deviation of the performance metrics across the 10 splits. The mean values provided an estimate of the models' average performance, while the standard deviation indicated the degree of variation in their performance across the splits. These statistics offered insights into the stability and reliability of the model's predictions.

### 4.2. Model Training Evaluation

1. DistilBERT: The DistilBERT model was trained on the dataset using appropriate hyperparameters and optimization techniques. After training, the model achieved a high accuracy of 97.48% on the evaluation set of 5000 entries, indicating its ability to effectively classify tweets as indicative of depression or not. The cross-validation results for the DistilBERT model yielded a mean accuracy of 97.174% with a standard deviation of 0.3965. The loss value of 0.314 further demonstrates the model's capability to minimize errors during training and optimize its predictive performance. A lower loss value indicates a better fit of the model to the training data. The accuracy and loss curves, shown in Figures 13 and 14, provide valuable insights into the model's training progress. The accuracy curve demonstrates how the model's accuracy improved over successive epochs, while the loss curve depicts the decreasing trend in the model's loss during training. These curves reflect the model's ability to learn from the training data and make accurate predictions.

    Throughout the training process, the accuracy curve steadily increased, indicating the model's ability to capture patterns and make more precise predictions. Simultaneously, the loss curve consistently decreased, indicating that the model successfully minimized errors and improved its overall performance. The confusion matrix is shown in Table 3 below.

**Table 3.** Confusion matrix for DistilBERT.

|  | **Actual Depression** | **Actual Non-Depression** |
| --- | --- | --- |
| Predicted Depression | 48.28% | 1.18% |
| Predicted Non-Depression | 1.68% | 48.86% |

2. SqueezeBERT: The SqueezeBERT model was trained on the dataset using the selected methodology. The accuracy metric indicates the proportion of correctly classified instances out of the total number of instances. In this case, the SqueezeBERT model achieved an accuracy rate of 95.48%, demonstrating its ability to effectively classify tweets as indicative of depression or not.

    The model demonstrated strong performance in cross-validation with a mean accuracy of 95.598% and a low standard deviation of 0.4416. This indicates the model's

consistency and reliability in accurately detecting patterns associated with depression. The loss value represents the error or discrepancy between the predicted outputs and the true labels. A lower loss value indicates better convergence and model performance. For the SqueezeBERT model, a loss of 0.285 was attained, suggesting a good fit of the model to the training data.

To visualize the training progress and model performance over epochs, the history curves of accuracy and loss were plotted in Figures 15 and 16. The accuracy curve depicts the increasing trend as the model learns to make more accurate predictions over time. Conversely, the loss curve shows a decreasing trend, indicating the diminishing error and improved convergence of the model. The confusion matrix is displayed in Table 4 below.

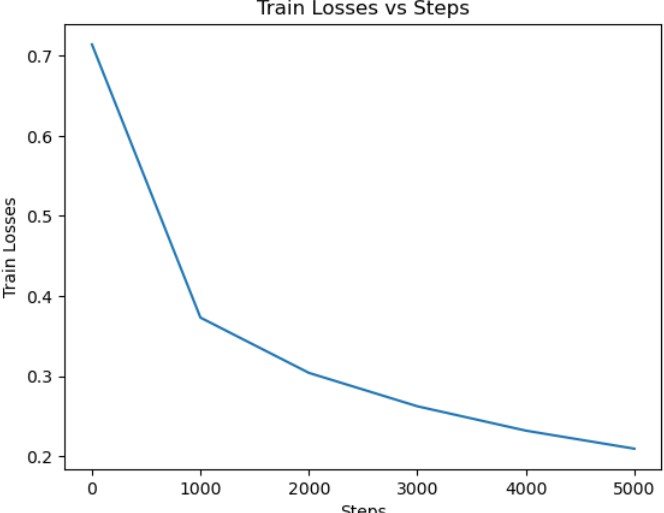

**Figure 13.** DistilBERT loss curve.

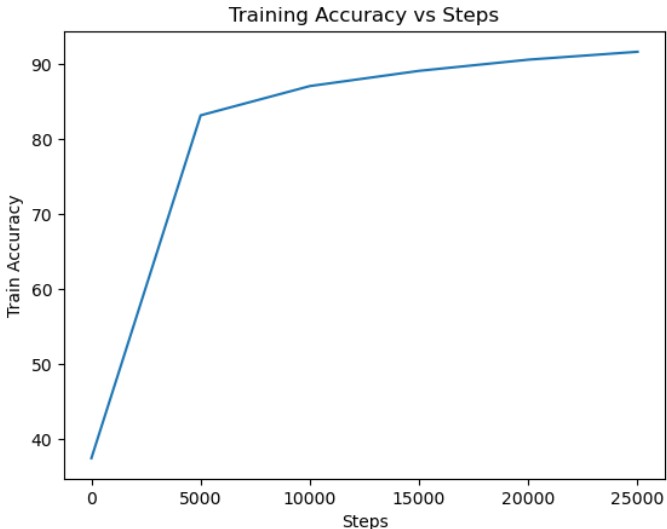

**Figure 14.** DsitilBERT accuracy curve.

**Table 4.** Confusion matrix for SqueezeBERT.

|  | **Actual Depression** | **Actual Non-Depression** |
| --- | --- | --- |
| Predicted Depression | 46.64% | 3.24% |
| Predicted Non-Depression | 1.28% | 48.84% |

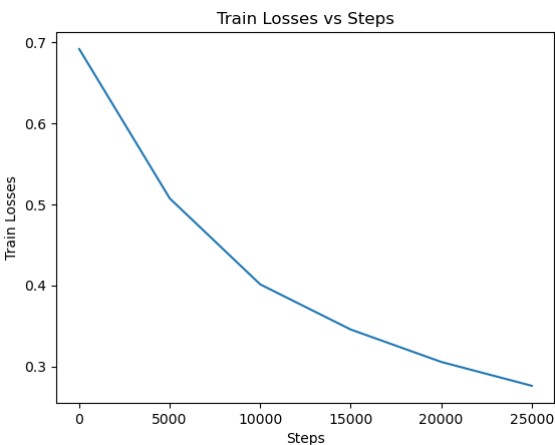

**Figure 15.** SqueezeBERT loss curve.

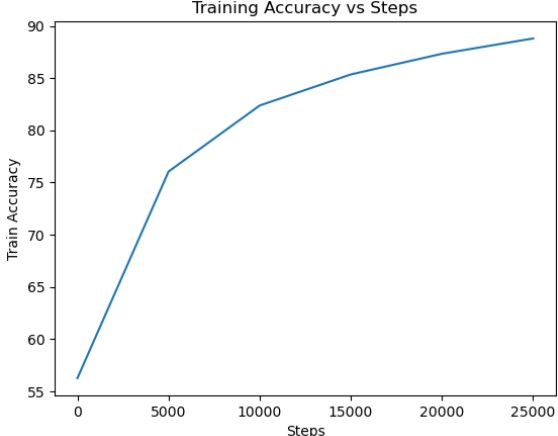

**Figure 16.** SqueezeBERT accuracy curve.

3. Logistic Regression: Upon evaluation, the logistic regression model achieved an accuracy of 0.97, indicating that it correctly classified 97% of the tweets. The precision of the model was found to be 0.972, which implies that 97% of the tweets predicted as indicative of depression were indeed depressive. The recall value of 0.967 suggests that the model identified 97% of the actual depressive tweets correctly.

   The F1 score, a measure that balances precision and recall, was calculated to be 0.97. This score indicates the overall effectiveness of the model in identifying depressive tweets. Additionally, the confusion matrix in Table 5 visually demonstrates the model's ability to discriminate between depressive and non-depressive tweets.

   The logistic regression model demonstrated strong performance during cross-validation, achieving an average accuracy of 93.111% with a standard deviation of 0.0016.

**Table 5.** Confusion matrix for logistic regression.

| | Actual Depression | Actual Non-Depression |
|---|---|---|
| Predicted Depression | 48.76% | 1.34% |
| Predicted Non-Depression | 1.64% | 48.26% |

4. Bernoulli Naive Bayes: The evaluation results of the Bernoulli Naive Bayes model indicated promising performance in detecting depression from user tweets. The model achieved an accuracy of 90.09%, a precision of 90.12%, a recall of 90.01%, and an F1 score of 90.07%.

The high accuracy suggests that the model was able to make correct predictions for a significant proportion of the instances in the evaluation set. The precision value indicates a strong ability to correctly identify tweets indicative of depression, while the recall value demonstrates the model's capability to capture a large portion of the actual positive instances.

The Bernoulli Naive Bayes model was evaluated using a 10-fold cross-validation approach. The mean accuracy across the 10 folds was calculated as 90% with a standard deviation of 0.0091. The confusion matrix is shown in Table 6 below.

**Table 6.** Confusion matrix for Bernoulli N.B.

|  | **Actual Depression** | **Actual Non-Depression** |
|---|---|---|
| Predicted Depression | 45.20% | 4.92% |
| Predicted Non-Depression | 4.98% | 44.90% |

5. Random Forests: The random forests model achieved an accuracy of 0.949 on the evaluation set, indicating that 95% of the instances were correctly classified. The precision of the model was measured at 0.964, reflecting the proportion of correctly predicted positive instances (tweets indicating depression) out of all predicted positive instances at 96%. The recall (or sensitivity) of the model was 0.933, indicating the proportion of correctly predicted positive instances out of all actual positive instances was at 93%.

   The F1 score, which balances precision and recall, was calculated as 95%. To gain further insights into the model's performance, a confusion matrix was constructed. The confusion matrix, as seen in Table 7, provides a detailed breakdown of the model's predictions by comparing them to the actual class labels.

   The random forests model demonstrated strong performance in the cross-validation process, achieving a mean accuracy of 92.7% with a low standard deviation of 0.0017. This indicates consistent and reliable predictions across the different folds of the dataset, highlighting the model's robustness in detecting the target variable.

**Table 7.** Confusion matrix for random forests.

|  | **Actual Depression** | **Actual Non-Depression** |
|---|---|---|
| Predicted Depression | 48.20% | 1.74% |
| Predicted Non-Depression | 3.37% | 46.69% |

6. RoBERTa: The RoBERTa model was trained on about 50% of the dataset (300,000 entries) using the aforementioned methodology because of the constraint of compute power. After training, the model's performance was evaluated using various metrics to assess its effectiveness in detecting depression from user tweets. Additionally, the confusion matrix in Table 8 visually demonstrates the model's ability to discriminate between depressive and non-depressive tweets. The cross-validation result for the RoBERTa model yielded a mean accuracy of 96.842% with a standard deviation of 0.847, showcasing its robust and consistent performance in detecting the desired outcome.

   The evaluation results demonstrated the promising performance of the RoBERTa model. It achieved an accuracy of 0.981, indicating that 98% of the tweets were correctly classified as indicative of depression or not with a loss of 0.2016. The precision, recall, and F1 scores were measured at 0.99, 0.99, and 0.98, respectively, indicating a good balance between correctly predicted positive instances and overall performance. The loss and accuracy curves showing the history of the training performance for the model are shown in Figures 17 and 18.

**Table 8.** Confusion matrix for RoBERTa.

|  | **Actual Depression** | **Actual Non-Depression** |
| --- | --- | --- |
| Predicted Depression | 49.08% | 1.11% |
| Predicted Non-Depression | 0.56% | 49.25% |

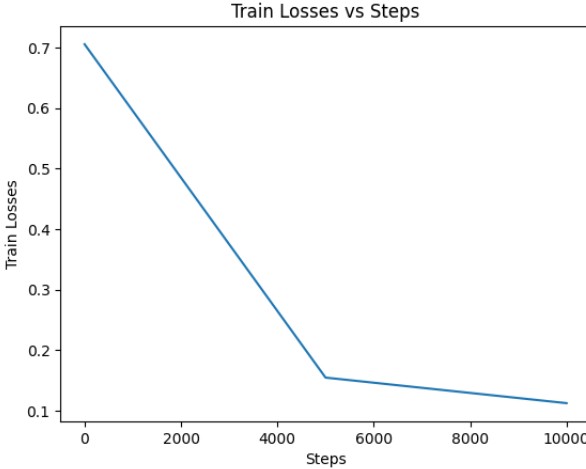

**Figure 17.** RoBERTa loss curve.

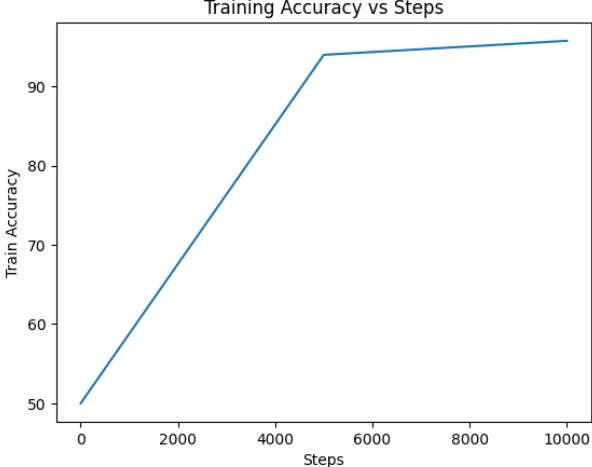

**Figure 18.** RoBERTa accuracy curve.

7.    DeBERTa: The DeBERTa model was also trained on about 50% of the dataset using the chosen methodology. After training, the model achieved an accuracy of 98% on the evaluation set. This indicates that the model accurately predicted depression and non-depression tweets with a high level of precision. Also, the confusion matrix, as seen in Table 9, provides a detailed breakdown of the model's predictions by comparing them to the actual class labels.

The DeBERTa model demonstrated strong performance during cross-validation, achieving an average accuracy of 95.688% with a low standard deviation of 0.504. This indicates consistent and reliable predictions across different subsets of the data, highlighting the model's robustness in detecting the target variable.

Furthermore, the loss value obtained for the DeBERTa model was 0.3438. The loss function measures the discrepancy between the predicted and actual values during training with lower values indicating a better fit of the model to the data. The training process was monitored by tracking the history curves of accuracy and loss. The accuracy curve depicts the improvement of the model's accuracy over each

training epoch, while the loss curve represents the decrease in the loss value over the same period. These curves provide insights into the model's learning progress and convergence, as shown in Figures 19 and 20.

The results and evaluation of the various models in this study demonstrated their effectiveness in detecting depression from user tweets. The logistic regression, Bernoulli Naive Bayes, and random forests models showcased strong performance with high accuracy and precise predictions. Moreover, the advanced transformer-based models, DistilBERT, SqueezeBERT, RoBERTa, and DeBERTa, exhibited exceptional accuracy and robustness, showcasing their potential for accurate and nuanced depression detection. These findings highlight the significance of leveraging machine learning algorithms and advanced models in analyzing social media data for mental health monitoring and support.

**Table 9.** Confusion matrix for DeBERTa.

|  | **Actual Depression** | **Actual Non-Depression** |
|---|---|---|
| Predicted Depression | 49.01% | 1.18% |
| Predicted Non-Depression | 4.11% | 49.43% |

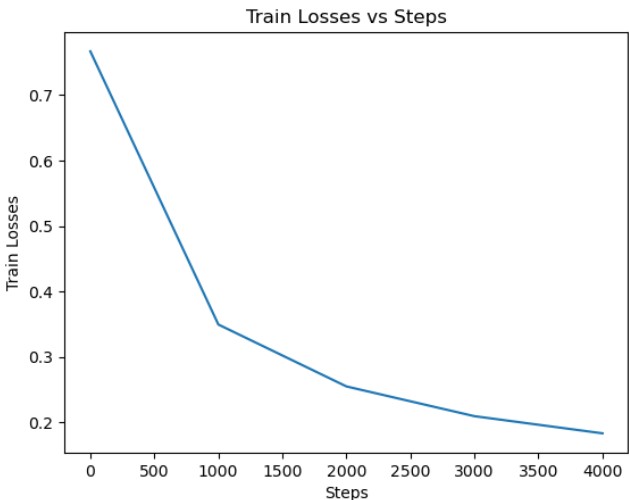

**Figure 19.** DeBERTa loss curve.

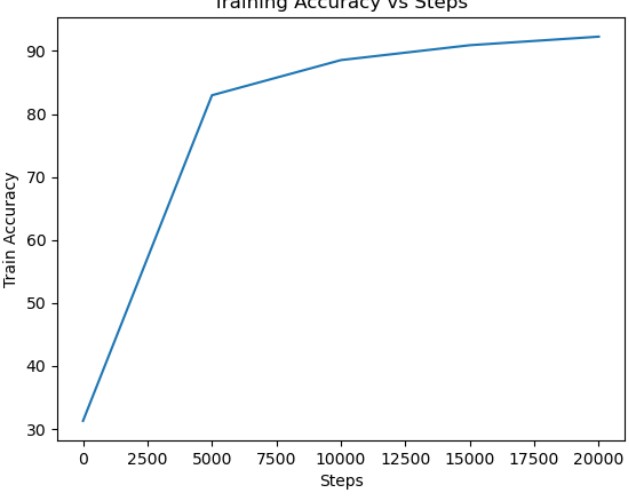

**Figure 20.** DeBERTa accuracy curve.

## 5. Conclusions

In conclusion, this paper presented a comprehensive approach for detecting depression from user tweets using machine learning techniques. The study utilized a large dataset consisting of 632,000 tweets, and a series of preprocessing steps were applied, including feature selection and data cleaning. By employing logistic regression, Bernoulli Naive Bayes, random forests, DistilBERT, SqueezeBERT, RoBERTa, and DeBERTa models, we achieved promising results (Table 10) in accurately identifying tweets indicative of depression.

**Table 10.** Evaluation metrics for the models.

| Model/Measure | Accuracy | Precision | Recall | F1-Score |
|---|---|---|---|---|
| Random Forest | 94.9% | 96.4% | 93.3% | 95.0% |
| Bernoulli N.B. | 90.1% | 90.1% | 90.0% | 90.1% |
| Logistic Regression | 97.0% | 97.2% | 96.7% | 97.0% |
| RoBERTa | 98.0% | 98.0% | 99.0% | 98.0% |
| DeBERTa | 98.0% | 98.0% | 98.0% | 98.0% |
| DistilBERT | 97.0% | 98.0% | 98.0% | 97.0% |
| SqueezeBERT | 95.0% | 97.0% | 97.0% | 96.0% |

The evaluation of the models revealed the effectiveness and robustness of the methodology. The RoBERTa model demonstrated exceptional performance, achieving a mean accuracy of 0.97 (across 10 cross-validation folds) and an accuracy of 98.1% on the evaluation set. These high accuracies indicate the model's ability to accurately distinguish between tweets associated with depression and those that are not.

The findings of this study have significant implications for mental health analysis using social media data. By leveraging machine learning techniques, we have demonstrated the potential for the early detection and monitoring of depression through user tweets. This approach has the advantage of being scalable and accessible, as social media platforms provide a vast amount of user-generated data that can be leveraged for mental health insights.

It is important to note that this study is just the beginning in the field of depression detection from user tweets. Further research can explore additional feature engineering techniques, investigate other machine learning algorithms, and explore the transferability of models to different social media platforms and languages. Moreover, combining textual information with other contextual data, such as user demographics or temporal patterns, can enhance the accuracy and reliability of depression detection models.

**Author Contributions:** Conceptualization, B.G.B. and Q.L.; Methodology, B.G.B. and Q.L.; Software, B.G.B.; Validation, B.G.B. and Q.L.; Formal analysis, B.G.B.; Investigation, B.G.B.; Resources, B.G.B. and Q.L.; Data curation, B.G.B. and Q.L.; Writing—original draft, B.G.B. and Q.L.; Writing—review & editing, B.G.B.; Visualization, B.G.B.; Supervision, B.G.B.; Project administration, B.G.B.; Funding acquisition, Q.L. All authors have read and agreed to the published version of the manuscript.

**Funding:** The research received no funding.

**Data Availability Statement:** Publicly available datasets were analyzed in this study. This data can be found here: http://help.sentiment140.com/for-students (accessed on 2 October 2023).

**Conflicts of Interest:** The authors declare no conflict of interest.

## Abbreviations

The following abbreviations are used in this article:

| | |
|---|---|
| BERT | Bidirectional Encoder Representations from Transformers |
| DistilBERT | Distilled Bidirectional Encoder Representations from Transformers |
| SqueezeBERT | Squeeze (Compressed) Bidirectional Encoder Representations from Transformers |
| RoBERTa | Robustly Optimized BERT Pretraining Approach |
| DeBERTa | Decoding-enhanced BERT with disentangled attention |
| SVM | Support Vector Machine |
| TF-IDF | Term Frequency-Inverse Document Frequency |
| ML | machine learning |
| AI | artificial intelligence |

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
