# Peer review of "Deep Learning-Based Depression Detection from Social Media: Comparative Evaluation of ML and Transformer Techniques"

_electronics, doi:10.3390/electronics12214396_

Round 1
Reviewer 1 Report
This manuscript exhibits promising research on depression detection from social media using the LLM model. However, to enhance the quality and impact of this manuscript, the authors should address the mentioned points, providing clear explanations, thorough evaluations, and comprehensive visualizations.
1. The authors did not clearly specify the source of the dataset used in this research.
2. The authors mentioned employing the Bert model in their research but did not sufficiently highlight the novelty of their algorithms.
3. To evaluate their predicted results, the authors should consider employing a well-established evaluation metric specific to depression detection or sentiment analysis.
4. The paper lacks clarity on the methodology used to obtain positive samples for the depression detection task.
5. The authors are encouraged to include visualizations of the confusion matrix and ROC curves in their paper.
6. The review suggests that the authors include detailed information about their computer settings, such as the CUDA version and GPU type.
Author Response
- The about dataset section (3.1) now clearly explains the source of the dataset and the study it was obtained and why it was used for this study.
- The methodology subsection “Model Training” explains the hyperparameters useed for the ML models and the model architecture used for the BERT finetuning is also specified
- The evaluation section clearly specifies why the chosen metrics were picked for the depression detection.
- System details are specified in methodology
Reviewer 2 Report
Attached , you can find my comments !

Author Response
- Introduction section includes more subsection and literature review section was extracted from it as specified in the review
- More descriptive details about the source of the dataset and why it was used has been specified. Details of how the dataset was used for the purpose of depression detection has also been clearly specified.
- In subsection 3.4 (formerly 2.4) more details about the model framework has been provided before the image.
- Discussion section renamed to conclusion.
- 4 more books added as literature reference (3 books from the suggestion of the reviewer)
Reviewer 3 Report
Depression is a prevalent mental health condition that affects a substantial number of individuals worldwide. This paper studied depression detection problem from social media using machine learning techniques. The methodology of the paper encompasses data collection, preprocessing, feature extraction, and the selection and training of machine learning algorithms. The study utilized a large dataset and the performance evaluation revealed the effectiveness and robustness of the methodology.
Although the topic is interesting, there are shortcomings in the innovative refinement of the paper. The research demonstrates the effectiveness of machine learning and advanced transformer-based models in leveraging social media data for health analysis, but similar conclusions have already been drawn from related work. The experiments require comparison and analysis with other methods.
Author Response
- The study has been refined to effectively show research gaps that it’s filling and the methodology has been refined to show how it fills those gaps
Reviewer 4 Report
Notes in the attachment.

Author Response
- Title of the article has been adjusted to fit the content of the article
- More descriptive details about the source of the dataset and why it was used has been specified. Details of how the dataset was used for the purpose of depression detection has also been clearly specified.
- Definition of word cloud has been provided and the analytic diagram has also been explained
- Removed drawings and tables as beginnings of chapters and added some text before them instead
- The methodology subsection “Model Training” explains the hyperparameters useed for the ML models and the model architecture used for the BERT finetuning is also specified
- Loss function and optimizer equations removed
- Excessive details about train test split removed from the results section and evaluation section clearly specifies why the chosen metrics were picked for the purpose of depression detection
Reviewer 5 Report
Dear Authors,
Thanks for giving me a chance to read this manuscript, “Deep Learning-based Depression Detection from Social Media: Comparative Evaluation of LLM Techniques”. The current paper a comprehensive approach for depression detection from user tweets using machine learning techniques. The study utilizes a dataset of 632,000 tweets and employs data preprocessing, feature selection, and model training with logistic regression, Bernoulli Naive Bayes, random forests, DistilBERT, SqueezeBERT, RoBERTa, and DeBERTa models.
This is an interesting and significant topic in the field of digital health. However, there are minor issues in the current manuscript that should be addressed.
1. Method
· The most ambiguous concern for me is the missing method of actual depression detection. As authors mentioned, “The proposed system aims to leverage the sentiment140 dataset to detect depression based on the empirical evidence provided by studies in the literature review [5,8].” However, authors are strongly advised to explicitly indicate the way how you label the grounded truth, namely, actual depression in the table.
To sum up, I personally like this paper. However, the problems should be addressed in order to be further considered. Hope these suggestions help.
fairly okay
Author Response
- The about dataset section (3.1) now clearly explains the source of the dataset and the study it was obtained and why it was used for this study.
Round 2
Reviewer 3 Report
The paper presents a comprehensive approach for depression detection from user tweets using machine learning techniques. The findings of the paper offer valuable insights into the potential for early detection and monitoring of depression using online platforms.
The authors have made revisions according to the reviewer's requirements.
Reviewer 4 Report
Without comments.
Reviewer 5 Report
The authors have explained fairly well. It is sufficient to publish in the current form.
sufficient enough